# Feasibility and Effectiveness Studies with Oral Vaccination of Free-Roaming Dogs against Rabies in Thailand

**DOI:** 10.3390/v13040571

**Published:** 2021-03-29

**Authors:** Karoon Chanachai, Vilaiporn Wongphruksasoong, Ad Vos, Kansuda Leelahapongsathon, Ratanaporn Tangwangvivat, Onpawee Sagarasaeranee, Paisin Lekcharoen, Porathip Trinuson, Suwicha Kasemsuwan

**Affiliations:** 1USAID Regional Development Mission Asia (Former at the Department of Livestock Development, Thailand), Athenee Tower, 25th Floor, 63 Wireless Road, Lumpini, Patumwan, Bangkok 10330, Thailand; 2Department of Livestock Development, Ministry of Agriculture, Ratchathewi, Bangok 10400, Thailand; lhin_001@hotmail.com (V.W.); ammieloveu@hotmail.com (O.S.); porathip.trinuson@hotmail.com (P.T.); 3CEVA Innovation Center GmbH, Am Pharmapark, 06861 Dessau-Rosslau, Germany; 4Faculty of Veterinary Medicine, Kasetsart University, Kamphaeng Saen 73140, Thailand; fvetkul@ku.ac.th (K.L.); fvetswk@ku.ac.th (S.K.); 5Department of Disease Control, Ministry of Public Health, 88/21 Tiwanon Rd., Talard-Kwan, Nonthaburi 11000, Thailand; ratanaporn.tw@gmail.com; 6Faculty of Veterinary Science, Chulalongkorn University, 39 Henri-Dunant Rd., Wangmai, Pathumwan, Bangkok 10330, Thailand; dvm.starling@hotmail.com

**Keywords:** rabies, dog, oral vaccination, Thailand

## Abstract

(1) Background: Thailand has made significant progress in reducing the number of human and animal rabies cases. However, control and elimination of the last remaining pockets of dog-mediated rabies have shown to be burdensome, predominantly as a result of the large numbers of free-roaming dogs without an owner that cannot be restrained without special efforts and therefore remain unvaccinated. To reach these dogs, the feasibility, and benefits of oral rabies vaccination (ORV) as a complementary tool has been examined under field conditions. (2) Methods: ORV of dogs was tested in five study areas of four provinces in Thailand. In these areas, sites with free-roaming dogs were identified with the support of local municipal workers and dog caretakers. ORV teams visited each of five study areas and distributed rabies vaccine (SPBN GASGAS) in three bait formats that were offered to the dogs using a hand-out and retrieval model. The three bait types tested included: egg-flavored baits, egg-flavored baits pasted with commercially available cat liquid snack, and boiled-intestine baits. A dog offered a vaccine bait was considered vaccinated when the discarded sachet was perforated or if a dog chewed vaccine bait at least 5 times before it swallowed the bait, including the sachet. (3) Results: A total of 2444 free-roaming dogs considered inaccessible for parenteral vaccination were identified at 338 sites. As not all dogs were approachable, 79.0% were offered a bait; of these dogs, 91.6% accepted the bait and subsequently 83.0% were considered successfully vaccinated. (4) Conclusion: Overall, 65.6% of the free-roaming dogs at these sites were successfully vaccinated by the oral route. Such a significant increase of the vaccination coverage of the free-roaming dog population could interrupt the rabies transmission cycle and offers a unique opportunity to reach the goal to eliminate dog-mediated human rabies in Thailand by 2030.

## 1. Introduction

Rabies still kills tens of thousands of people every year worldwide, although highly effective vaccines to prevent these deaths are widely available [1]. More than 95% of these human rabies cases are acquired from the bite of an infected domestic dog [1]. Hence, eliminating dog-mediated human rabies could make a significant contribution in the reduction of the disease burden. The single most cost-effective method to reach this goal is by eliminating rabies at its source through dog vaccination [1]. Mass dog vaccination campaigns can confer herd immunity and successfully disrupt the rabies transmission cycle among this reservoir species. This approach has been implemented successfully in many countries all over the world [2]. Also, in Thailand significant progress has been made in the elimination of rabies. Human rabies cases per year decreased from 200–300 in early 1980s [3] to three in 2019 (available at http://www.boe.moph.go.th/boedb/surdata/index.php, accessed on 10 January 2021). Animal rabies also experienced the same trend; it was significantly reduced from 3–4000 confirmed cases per year during 1997–2000 to 380 cases in 2019 (available at http://www.thairabies.net/trn/ assessed on 20 March 2021). However, controlling and eliminating the last remaining rabies foci in dogs was more difficult than expected. Hence, the target year set to become dog-rabies free has been repositioned several times. In 2019, 87% of animal rabies in Thailand were reported in dogs (available at http://www.thairabies.net/trn/, accessed on 20 March 2021). Most of these dogs involve free-roaming dogs, with and without an owner, that never received a rabies vaccination. In Thailand, most of the free-roaming dogs are being fed and looked after by someone in the community (dog caretakers), but these caretakers have limited capabilities to capture or restrain these dogs. Therefore, these dogs cannot typically be vaccinated by more standard methods using parenteral vaccines during Mass Dog Vaccination (MDV) campaigns, which have been conducted annually during April–June by both methods, which are door-to-door and central point vaccination, depending on the preferences of local responsible authorities.

Oral vaccination of wildlife has been successfully used to control rabies in different wildlife species around the world and it has also been suggested for dogs that are inaccessible for parenteral vaccination [4]. We studied the feasibility and field effectiveness of the use of oral rabies vaccine targeting free-roaming dogs as a complementary tool to control rabies in Thailand. The study design consisted of 3 phases. The first phase was identifying the most appropriate bait for the target dog population in Thailand that is well accepted and facilitates the release of the vaccine in the oral cavity. Secondly, the vaccine candidate must not only meet minimum safety requirements as laid down by the World Organisation for Animal Health (OIE) [5] but also induce an adequate immune response after the consumption of a single bait. Finally, the feasibility of oral rabies vaccination (ORV) needs to be assessed during field trials in a variety of local settings, as suggested by the World Health Organization (WHO) [1]. The first phase study—a bait acceptance study—demonstrated that a large portion of the free-roaming dog population could be reached by offering the egg-flavored or boiled-intestine baits, while the egg-flavored bait was better suited for the release of the vaccine in the oral cavity [6]. The second phase study—a serology study—indicated that the selected vaccine candidate, SPBN GASGAS, induced a sustained and detectable immune response after oral administration in local dogs comparable to dogs receiving a commercially available parenteral vaccine [7]. The results of the third phase study—a feasibility and effectiveness study—are presented in this paper. The major objectives were to determine the benefits of ORV of dogs as a complementary tool to parenteral vaccination and to optimize the application of ORV in free-roaming dogs by identifying possible constraints and solutions for the integration of this method in MDV-campaigns under local settings.

## 2. Materials and Methods

Initially, 4 municipalities with a known problem restraining and vaccinating the stray dog population during traditional MDV campaigns and that were willing to participate in the project were selected: The four municipalities consist of two municipalities in Rayong province (Choeng Noen, Phe), one municipality in Phetchaburi province (Cha Um), and another in Nakhonsri Thammarat province (Thung Song) (Figure 1). The approximate dog population in each four municipalities was estimated at 2000–3000 owned dogs and 500 ownerless dogs. These estimates were based on previous MDV vaccination campaign results. The ORV field studies were carried out before (Thung Song) or shortly after (Choeng Noen, Phe, and Cha Um) a parenteral MDV-campaign in these areas. Another additional rural area in the eastern part of Thailand, Tapraya, was included in a later stage to determine ORV’s benefit in controlling an on-going rabies outbreak in free-roaming dogs. Field studies were conducted during March–August 2020.

Field studies were conducted using the SPBN GASGAS vaccine (Ceva Innovation Center GmbH, Dessau in Germany) based on previous serologic studies [7]. The 3rd generation vaccine virus, SPBN GASGAS, is a genetically engineered derivate of SAD L16, a cDNA clone of the 1st generation oral rabies virus vaccine SAD B19. SPBN GASGAS lacks the pseudogene (ψ) and for safety purposes it encodes the amino acids glutamic acid and serine at position 333 and 194 of the rabies virus glycoprotein, respectively. The modification at position 333 of the glycoprotein eliminated the residual pathogenicity observed in adult mice after intra cranial inoculation with SAD B19 and the modification at position 194 circumvented a potential compensatory mutation restoring the original residual pathogenicity. All three nucleotides at both positions were changed to reduce the risks for back mutations to its original form [8]. Furthermore, an additional identical altered glycoprotein was inserted to further enhance the safety profile of the vaccine virus [9].

A sachet filled with the liquid vaccine virus (3 mL, 10^8.2^ FFU/mL) was incorporated in two different bait types previously shown to be readily accepted by local free-roaming dogs in Thailand: an industrial manufactured egg-flavored bait (egg bait) and a locally produced intestine bait. As the study team observed that the acceptance of the egg bait could be further optimized, tuna- or chicken liver-flavored cat liquid snacks available in the local markets were pasted on one side of the outer surface of the bait immediately before offering the bait to the dog (egg+ bait), as seen in Figure 2.

Staff at the Department of Livestock Development (DLD), Ministry of Agriculture and Cooperatives, prepared the intestine baits by inserting the sachet with frozen vaccine into an 8–12 cm segment of boiled (pork) intestine. Vaccine baits were then immediately placed back in a freezer to prevent thawing of the vaccine virus. Prior to the ORV-campaign, an inventory was made with local municipality workers and dog caretakers on the locations where the free-roaming dogs could be found, followed by an estimation of the number of dogs present on these sites. At the beginning of the ORV-campaign, the vaccination team members received brief training on oral rabies vaccination, including vaccine bait handling, techniques for approaching dogs, methods for offering the vaccine bait, recording vaccine bait handling by individual dogs (duration, consumption, perforation and/or swallowing of sachet), interpreting effectiveness of vaccination attempt, and underscoring the importance of retrieving the discarded vaccine sachet after bait consumption. A hand-out and retrieve model was used to distribute vaccine baits to the targeted dog population [10]. Dogs considered inaccessible for parenteral vaccination were offered a bait. If the animal ran away or did not accept the bait, the bait was recollected by the vaccination team and offered to the next dog. Also, the (perforated) sachets discarded by the dogs after bait consumption were recollected by the vaccination teams. This way, unintentional human contacts with the vaccine virus could be reduced considerably. For planning purposes, it was assumed that the ORV campaign could be completed within one week in each municipality. Because the number of free-roaming dogs and sites differed among the five selected study areas, two to four vaccination teams were deployed for each ORV-campaign. In general, each team consisted of four members who were responsible for free-roaming dog tracking, vaccination, observing, and data recording. Only one bait type was used by a team during a particular day. In case of the egg bait, the team made a decision whether to add the cat liquid snack paste on the egg bait (egg+ bait) or not, if the dogs seemed to be difficult to access. Vaccine (baits) were stored at −20 °C (manufacturer and DLD) and transported using dry ice. Upon arrival at the field study areas, it was kept frozen in a standard style freezer with a −18 °C or lower freezing compartment. Vaccine baits were transferred to cool boxes overnight before field use, ensuring that baits had thawed before they were offered to the dogs. Baits unused at the end of the vaccination day were kept at refrigerator temperatures (4–8 °C) and offered to dogs the next day. The vaccine baits were used within 3 days after thawing. Oral rabies vaccinators received pre-exposure vaccination prior to field studies. Personnel handing the baits wore examination gloves. The recollected discarded sachets were disposed of as infective materials at the primary health care unit of the municipality.

Upon arrival at the pre-identified sites, dogs encountered were assessed to determine whether they were eligible for oral vaccination. The exclusion criteria for oral vaccination, judged by local people or vaccination team, were as follows: dogs able to be restrained for parenteral vaccination without special effort, puppies (<3 months old), and dogs with a disease or injury that might interfere with the induction of an immune response. 

A dog offered a vaccine bait was considered to be vaccinated when the discarded sachet was perforated or if a dog chewed vaccine bait at least 5 times before it swallowed the bait, including the sachet. For every dog offered a bait, bait handling (bait acceptance, bait consumption, fate of sachet—perforated or not, discarded or not) together with dog demographic data (such as sex, age, size, single or together with other dogs) was recorded in a hard copy questionnaire. Also, potential direct and indirect contacts with the vaccine virus involving humans and non-target species were documented. Furthermore, the site was characterized in certain habitat types such as village, public places, main roadside, temple, beach, school, etc. The number of free-roaming dogs at these sites were recorded together with the number of dogs offered a bait vaccine. Information of each site including numbers of free-roaming and vaccinated dogs, the time and duration spent at each site, and the geographical location of the site were recorded using the Epicollect 5 application [11].

The vaccination teams informed the local people present at the sites visited on ORV and the purpose of the study during the visit. Key focal persons were offered a leaflet containing information and contact details. After the campaign, follow-up calls with the local dog caretaker and other contact people available in each study area were made if any adverse event after vaccination including human and non-target species contacts with the vaccine virus had been reported. Also, meetings with key partners like ORV team members and municipality staff were conducted post the ORV-campaign to determine their perceived benefits and concerns associated with ORV of dogs. No ethical approval was needed for this study as animals were not forced in any way to accept the vaccine bait.

### Statistical Analysis

Statistical analysis was performed by chi-square test and a logistic regression model. In the logistic regression model, the dependent variable was determined as “vaccination success” (the discarded sachet was perforated or if a dog chewed vaccine bait at least 5 times before it swallowed the bait including the sachet). Independent variables were the study area (Choen Noen, Cha Um, Phe, Thong Song, or Tapraya), the person offering the vaccine bait (caretaker, volunteer, municipality, or study team staff), method of offering the bait to the dog (direct [hand-feeding], drop, or throw), dog’s social status during bait offering (single or together with other dogs), dog size (less than 10, 10–40, or >40 kg), dog sex (male or female), dog age (juvenile [<1 year] or adult [≥1 year]), and bait types (intestine-, egg−, or egg+ bait). Individual independent variables were initially screened for associations with vaccination success. All independent variables with *p* < 0.2 for at least one level in the univariable analyses were submitted to a multivariate model. The multivariate model was created using a backward manual stepwise process logistic regression. Factors with *p* ≤ 0.05 in the multivariate analysis were included into the final model. Statistical analysis was conducted using R version 4.0.2.

## 3. Results

A total of 338 sites with an estimated 2444 free-roaming dogs in the five study areas were identified. The habitat-type of the sites where the dogs were offered baits is shown in Figure 3. The free-roaming dogs were most frequently found in four primary habitats characterized as village/public space (35.5%, 120/338), temple (14.5%, 49/338), beach (11.0%, 37/338), and main roadside (11.0%, 37/338), respectively. 

We attempted offering baits to 1930 dogs. Most free-roaming dogs at the sites were observed in groups (90.8%); group size ranged from 2 to 47 dogs. The bait was offered directly to 12.1% of the dogs (231/1910). In some cases, the bait was dropped in front of the dog when passing by (45.5%, 869/1910) and in other cases the dog could not be approached directly and the bait was tossed/thrown to the animals (42.4%, 810/1910). No information on the method of delivery was available for 20 dogs. Most baits were offered by municipality or DLD staff (74.6%, 1397/1873), followed by dog caretakers (19.9%, 372/1873) and animal/public health volunteers (5.6%, 104/1873). For 57 dogs, it was not recorded who offered the bait. The raw data can be found in the Appendix A.

The most frequent bait offered to the dogs was the intestine bait (68.1%, 1314/1930), followed by egg bait (17.5%, 338/1930) and the egg+ bait (14.4%, 278/1930). As observed during previous bait acceptance studies (5,16), dogs were most interested in the intestine baits, but the egg baits were better suited for delivery (release) of the vaccine in the oral cavity (Table 1). Dogs were significantly more interested in the intestine bait than in the egg baits; the rates were 92.9% versus 87.3%, respectively (Chi² = 10.10, df = 1, *p* = 0.002). Adding the cat liquid snack paste on the egg bait (egg+ bait) improved bait acceptance significantly; the rates were 92.8% versus 87.3% (Chi² = 4.34, df = 1, *p* = 0.04), but it did not influence the swallowing rate of the sachet significantly. The sachet inside the intestine bait was significantly more often swallowed than the sachet inside the egg baits; the rates were 80.0% versus 32.2%, respectively (Chi² = 242.39, df = 1, *p* < 0.0001). Also, the intestine bait was significantly more rapidly consumed than the other two bait types; 42.5% of the dogs ate the intestine bait within 10 s (Chi² = 95.43, df = 2, *p* < 0.0001). The vaccination rate was not significantly different between intestine and egg bait types because the sachet within the more attractive intestine bait was swallowed more frequently and also the chewing time was reduced compared to the egg baits, affecting the release of the vaccine in the oral cavity negatively. Pasting cat liquid snack on egg bait (egg+ bait) improved the vaccination success significantly compared to the standard egg bait (Chi² = 3.894, df = 1, *p* = 0.048).

No detailed data describing the free-roaming dog population in each municipality was available. Vaccination coverage was evaluated by using the estimated number of dogs on each site identified. The vaccination coverage obtained in the different study areas is summarized in Table 2. Overall, 65.6% of free-roaming dogs were successfully vaccinated by offering a vaccine bait. We failed to vaccinate 34.4% of the free-roaming dogs previously located at the sites, as some attempts were not successful (13.4%, e.g., sachet not perforated or bait immediately swallowed) and we were not able to approach and offer a bait to the free-roaming dogs that were previously identified (21.0%). Dog caretaker or focal people in each site informed us that the time most of the free-roaming dogs were present at the sites was during 16:00–19:00 (46.6%, 81 sites in 174 sites) and 05:00–07:00 (19.5%, 34 sites in 174 sites). The activities of the vaccination teams overlapped only to a limited extent with these periods, 29.3% (99 sites in 338 sites) during 16:00–19:00 and 5.9%, (20 sites in 338 sites) during 05:00–07:00. Hence, not all dogs were present at the selected sites during the visits of the ORV-teams.

Most dogs (92.2%) consumed only one bait; however with larger group sizes, sometimes a dog would consume more than one bait, as it would grab baits offered to other group members. If multiple baits were consumed by an individual dog, most of the time it involved 1 (6.7%) or 2 (1.3%) additional baits; one dog consumed 6 baits.

Independent variables included in the final model determining oral rabies vaccination success were study area, dog’s social status when bait was offered, bait distributor, and bait type (Table 3). Vaccine baits distributed to dogs in groups (OR = 1.86, single as reference) and local staff or study team members as bait distributors (OR = 1.64, dog caretaker as reference) increased oral rabies vaccination success. In addition, egg+ bait increased oral rabies vaccination success by 1.76 times compared to intestine bait. Oral rabies vaccination success was higher in Choen Noen and Tapraya by 1.89 and 4.73, respectively compared to Cha Um. The other variables (method of offering the bait as well as dog size, age and sex) were excluded during univariable analysis or the backward manual stepwise process.

No non-target animal species consumed or contacted a vaccine bait. A total of 6 human contacts with the vaccine virus were reported. These contacts were a result of notwearing (intact) gloves during vaccine bait delivery (*n* = 4) and during the collection of discarded vaccine sachet (*n* = 2). No adverse effects were reported during the follow-up call to dog caretakers, municipality staff, and local contacts at each study site.

## 4. Discussion

This study demonstrates that a high percentage of both bait acceptance and vaccination coverage can be achieved in free-roaming dogs using the oral bait handout and retrieve method. As the free-roaming dog population plays a key role in the transmission of rabies among dogs [4], it is essential to target these animals during vaccination campaigns. Most free-roaming dogs cannot be easily handled and restrained for parenteral vaccination, limiting the ability of MDV-campaigns to achieve vaccination coverages required to eliminate the rabies virus. Advanced vaccination methods such as capture-vaccinate-release (CVR) have been incorporated in many MDV-campaigns [12,13,14], but several shortcomings have been identified. First, the CVR-method is extremely expensive, as it is both time and labor intensive. Secondly, it can cause potentially harmful and dangerous situations not only for the dogs, but also for the vaccinators and bystanders. Importantly, CVR becomes less effective and efficient overtime as dogs become more difficult to catch during subsequent CVR-campaigns [15,16]. As a result, oral vaccination seems to be a promising alternative for reaching these free-roaming dogs that are not accessible for parenteral vaccination without special effort. However, in contrast to parenteral vaccination, it is often difficult to assess if the animal is successfully vaccinated. Oral bait-based vaccines carry inherent uncertainty—even when the sachet has been perforated, it cannot be assumed that a sufficient amount of vaccine has been taken up by the tonsils or mucous membrane in the oral cavity; this is a prerequisite to induce a protective immune response. For example, vaccine released from the sachet can be spilled on the ground or swallowed rapidly and subsequently lose its immunogenic potential in the gastro-intestinal tract. However, our second phase study indicated that if a dog perforates the sachet as indicated by chewing on the bait, an adequate immune response will follow as shown in the serology in shelter dogs offered an intestine bait. The antibody response was comparable to parenteral vaccination 1 year after vaccination [7]. Hence, in this field study, it was decided to refrain from collecting blood samples from the dogs to confirm successful vaccination, as this was beyond the scope of the study to determine the feasibility and effectiveness of ORV in the free-roaming dog population.

Besides a locally made intestine bait, an industrially manufactured egg bait was used in this field study. Although it has been suggested that a universal well accepted bait may not exist for dogs [17], the egg bait was previously shown to be well received by dogs in different settings, including Thailand [6,18,19]. Adding a cat snack (paste) on the egg bait increased the bait acceptance and vaccination success in the present study, indicating that local available products can optimize the local bait acceptance of a mass-produced commercial bait without much effort. As the oral vaccine bait has to be kept at cold temperature, the smell of the cold vaccine bait may be less intense and thus attractive to the dogs. Adding paste on egg bait increased olfactory attractiveness and accelerated bait acceptance when offered to the dogs. This egg+ bait was predominantly offered to dogs considered difficult to reach. Hence, this selection bias may have resulted in an underestimation of the true acceptance and subsequent vaccine success rate of this bait compared to the other two bait types. Although the intestine bait was more readily accepted by the dogs than the egg baits, the latter were more efficient in releasing the vaccine in the oral cavity; this was a prerequisite for a successful vaccination attempt [20]. For example, the intestine bait was often swallowed within 10 s, including the sachet containing the vaccine. Hence, the liquid vaccine was most likely not released in the oral cavity and subsequently the dog was not considered vaccinated. Another variable that had a positive effect on vaccination success was if the targeted dog was offered a bait when conspecifics were nearby. This result is consistent with our field observation that solitary dogs were more selective in accepting the bait than dogs encountered in groups. Also, due to baiting experience and/or local knowledge of the dog population vaccination success was increased if bait distribution was carried out either by local staff or study team members. Local staff and study team members had the advantage of gaining experience throughout the research project in all five study sites, while individual dog caretakers were only involved in bait distribution at specific sites. Association between study areas and vaccination success indicates that there might be other independent variables influencing vaccination success which need to be explored further.

An issue often raised in the context of ORV of dogs is the lack of available methods for marking dogs for identification. Free-roaming dogs can be encountered on different occasions during a campaign and thus be offered a vaccine bait multiple times. Oral vaccination using the hand-out and retrieve model is by definition only applied for dogs considered not accessible; it is not possible to collar or otherwise mark the animals using methods that involve direct contact. Remotely marking the animals for example by paint-spraying is problematic and can give unreliable results [21,22]. In contrast to other ORV field studies where a systematic coverage of the area was conducted and all inaccessible dogs encountered were offered a bait, here prior to the campaign, the locations of the free-roaming dogs were assessed with the support of local staff and/or dog caretakers. Subsequently, these sites were visited, and dogs that were present were offered a vaccine bait. As these dogs often receive food on a regular basis by community members or dog caretakers at these sites, the individual dogs encountered were habituated to certain persons providing food and subsequently had strong fidelity to specific sites. Therefore, the chances that the vaccinators would come across the same dogs at other sites visited was reduced. ORV teams were asked to visit each site no more than two times to avoid double baiting dogs previously vaccinated. One major challenge to the site-specific vaccine bait handout model involves the size of the local population of dogs and the behavior of the dogs on site. Larger groups of dogs made it more difficult for baiters to reach the individual animals as some dominant dogs would succeed in consuming multiple baits. The involvement of the local dog caretakers offers many advantages. They have an excellent knowledge not only on the locations but also on the number of dogs and their accessibility to facilitate baiting. Additionally, the presence of the dog caretakers makes the dogs more approachable and generally helps encourage the dog to accept the bait being offered. The involvement of the dog caretakers ensures local support, which is essential for the support and implementation of the campaign, as well as follow-up and post-campaign monitoring.

Overall, 79.0% and 65.6% of the free-roaming dog population inaccessible for parenteral vaccination at the selected sites could be offered a bait and were subsequently considered successfully vaccinated. This significant increase of vaccination coverage clearly demonstrated the benefit of ORV in controlling rabies in free-roaming dogs, as these dogs had no known history of vaccination. Vaccination teams involved with this study received limited training and had limited experience approaching free-roaming dogs. Therefore, the vaccination coverage and efficiency will most likely increase during subsequent ORV campaigns as the teams gain experience and take an adaptive management approach to improving ORV campaigns. Local volunteers will likely continue to assist and refine the process for identifying areas for accessing free-roaming dogs and selecting the optimal time to visit these sites. Evidence for such improvements was already observed during these field studies. Such progress in practical skills as well as optimized collaboration with local stakeholders will be required to achieve and sustain a 70% vaccination coverage as recommended by WHO in the most efficient way [1]. No detailed cost-analysis has been conducted in this study as it has been shown previously in other studies that including ORV of dogs is cost-beneficial compared to other vaccination strategies when applied appropriately [21,23]. Furthermore, it must be examined whether targeting selected sites is sufficient to control rabies or if a more systematic approach at the landscape scale is needed to ensure the campaigns are reaching free-roaming dogs that may not be regularly visiting the sites identified for this study.

No adverse events were reported afterwards but several (potential) direct and indirect human contacts with the vaccine virus were observed during the campaign. The oral vaccine used in this study has a favorable safety profile and the hand-out and retrieve model used to distribute the baits limits the risk associated with unintentional human contacts with the vaccine virus to very low levels [24,25,26,27]. However, avoiding direct contact with vaccines during vaccination and collecting discarded sachet must be continuously emphasized throughout the ORV campaign, as the vaccinators may become less vigilant with time. During the post-campaign meeting with the team members, the use of oral rabies vaccine baits for inaccessible free-roaming dogs was strongly supported by the participants. Concerns identified regarding vaccine safety and the issue of “over vaccinating” dogs that were previously vaccinated by the parenteral route were considered negligible as these concerns could largely be prevented by raising public awareness and enhanced training of the baiters. Communication with the local community could also circumvent misperception perceived by the public on ORV (“poisoning campaigns of free-roaming dogs using baits”). Furthermore, we also recommended conducting the ORV-campaigns shortly after the MDV campaigns that are commonly conducted in April or May in Thailand. During these MDV-campaigns, owners should place a collar with a vaccination tag around the neck of their vaccinated dog. This way the ORV-team can easily differentiate dogs roaming freely in public areas but recently vaccinated by the parenteral route from non-vaccinated free-roaming dogs. Limited surveys should be conducted during MDV campaigns to identify free-roaming dog sites and population estimates of dogs present. Subsequently, the ORV team can estimate workloads and set the appropriate arrangements for oral vaccination (e.g., number of teams, duration of the campaign, local focal points [municipality, dog caretaker, and dog tracker], and targeted number of baits needed). Central and regional laboratories where a deep freezer is available, should be contacted to ensure the appropriate cold-chain. Finally, the commercial egg bait was recommended during the post campaign meeting in order to avoid the risk of interrupting the cold-chain during preparation of the local-made intestine vaccine bait.

## 5. Conclusions

This first large-scale field study in Thailand demonstrated that ORV of dogs could contribute significantly, improving the vaccination coverage of the free-roaming dogs considered inaccessible for parenteral vaccination. The protocol established during these field studies showed the potential to integrate ORV of dogs in existing dog rabies control programs. For reaching targeted vaccination levels, the approachability of the free-roaming dogs was crucial. This was only feasible through the cooperation and direct involvement of local authorities, community members, and dog caretakers. Hence, their support needs to be assured prior to implementation of the campaign. Our findings suggest that the use of ORV for free-roaming dogs, especially in areas with residual rabies foci in Thailand in the coming years, may contribute to achieving the goal of national elimination of dog-mediated rabies.

## Figures and Tables

**Figure 1 viruses-13-00571-f001:**
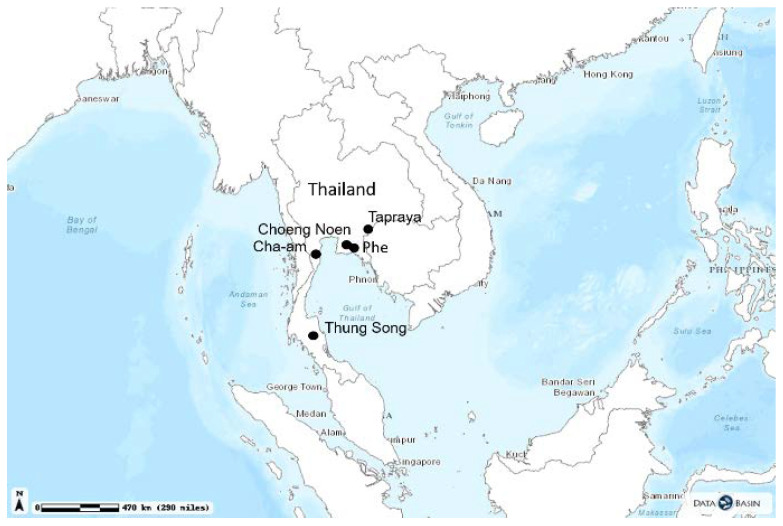
The location of the 5 study areas; (1) Choen Noen, (2) Cha Um, (3) Phe, (4) Thung Song, and (5) Tapraya.

**Figure 2 viruses-13-00571-f002:**
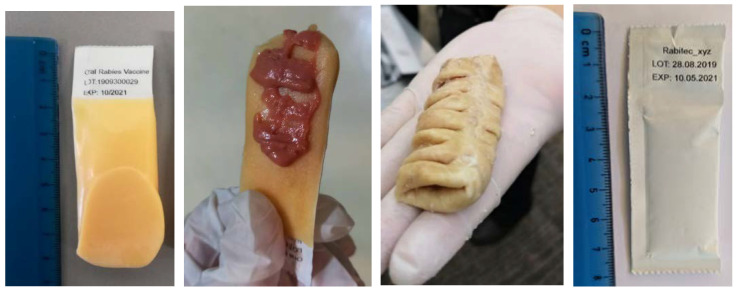
The egg bait (left), egg+ bait (middle left), and intestine bait (middle right) used in the field studies and vaccine sachet (right).

**Figure 3 viruses-13-00571-f003:**
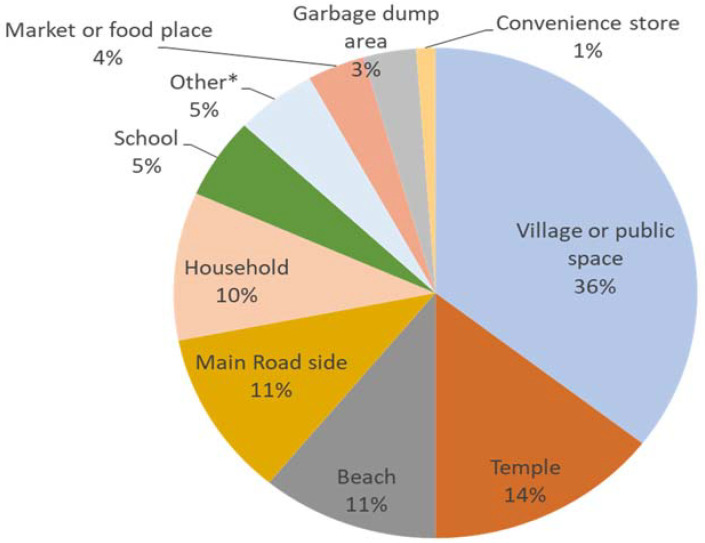
Frequency distribution of the habitat types of sites visited for oral vaccination in the 5 study areas (*n* = 338). * other: park, rubber tree area, fishpond, and construction site.

**Table 1 viruses-13-00571-t001:** Percentage of dogs interested in bait offered, dogs that chewed very shorty (<10 s), swallowed the sachet and considered vaccinated per bait type offered; intestine bait, egg bait, and egg+ bait (%—percentage, *n*—number of animals with positive result, N—total sample size).

Bait Type	No. of Dogs Offered a Vaccine Bait	No. of Dogs Interested in Bait (% [*n*/N]) *	Sachet Swallowed (% [*n*/N])	Chewing Time (<10 s) (% [*n*/N])	Vaccinated ** (% [*n*/N])
Intestine	1314	92.9 (1209/1302)	80.0 (929/1161)	42.5 (480/1130)	82.0 (995/1214)
Egg	338	87.3 (288/330)	32.2 (88/273)	24.0 (58/242)	83.6 (255/305)
Egg+	278	92.8 (256/276)	26.5 (65/245)	24.6 (60/244)	87.0 (235/270)
total	1930	91.9 (1753/1908)	64.4 (1082/1679)	37.0 (598/1616)	83.0 (1485/1789)

* some dogs had been excluded as they were scared away when the bait was offered; ** successful vaccination attempt is defined by perforated sachet or when dog chewed at least 5 times before swallowing bait and sachet.

**Table 2 viruses-13-00571-t002:** Achieved oral rabies vaccination coverage in the free-roaming dog population at the identified sites in 5 study areas.

Study Area	Nr. of Sites	Nr of Inaccessible Dogs	Dogs Approached (% [*n*/N])	Dogs Accepting the Bait & Successfully Vaccinated (% [m/M])	Vaccination Coverage Achieved (%) *
Choen Noen	59	488	77.5 (378/488)	88.1 (310/352)	68.2
Cha Um	59	789	71.7 (566/789)	79.5 (387/487)	57.0
Phe	112	564	86.5 (488/564)	81.9 (381/465)	70.9
Thong Song	77	456	87.7 (400/456)	81.2 (315/388)	71.2
Tapraya	31	147	66.7 (98/147)	94.9 (92/97)	63.2
Total	338	2444	79.0 (1930/2444)	83.0 (1485/1789)	65.6

* vaccination coverage achieved (%) was calculated as follows: 100 (*n*/N * m/M).

**Table 3 viruses-13-00571-t003:** Parameter estimates from the final multivariate model indicating factors associated with oral rabies vaccination success in free-roaming dogs in Thailand.

Factors	Coefficient	SE	Odds Ratio	95% CI	*p*-Value
Study area					
Choen Noen	0.64	0.23	1.89	1.22–2.99	0.006
Phe	0.18	0.18	1.19	0.84–1.71	0.329
Hong Song	0.21	0.19	1.23	0.85–1.80	0.275
Tapraya	1.56	0.48	4.73	2.04–13.80	0.001
Cha Um	Reference				
Social status of dog during vaccination					
Together with other dogs	0.62	0.19	1.86	1.27–2.68	0.001
Single	Reference				
Bait distributor					
Local staff or study team member	0.49	0.17	1.64	1.16–2.92	0.005
Volunteer	0.20	0.34	1.22	0.63–2.45	0.561
Dog caretaker	Reference				
Bait type					
Egg+	0.56	0.22	1.76	1.16–2.73	0.009
Egg	0.06	0.19	1.06	0.73–1.56	0.770
Intestine	Reference

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
