# Peer review of "Feasibility and Effectiveness Studies with Oral Vaccination of Free-Roaming Dogs against Rabies in Thailand"

_viruses, 2021, doi:10.3390/v13040571_

Round 1

Reviewer 1 Report

All in all, a very clearly and concisely written paper. The following points are raised as suggestions to improve the manuscript: 

Intro

Ll 44/45 provide reference

L 54 not sure why a median is presented here if a decreasing trend is to be shown

L 58 how is this different from the median shown in l 54?

Information missing: how were the MDV campaigns organized: door to door? Central point vaccination?

MM

More information on similar and distinctive characteristics of the municipalities with known problems reaching the stray dog population would be interesting. It could help identifying predictors of unaccessibility of dogs for future campaigns. However I see this is not the purpose of this paper and I leave it up to authors

Fig.2: Baits are pictured very well, but could a bait in a sachet be shown, too? Otherwise to the unexperienced reader it is hard to understand what is meant by the "sachet"

L 173 provide reference concerning the mentioned application

L 188 "person offering the vaccine"

Results

Fig.3 Garbage dump area

L211 what is meant by "directly" opposed to "dropping in front of dog"?

L219 ff it is not clear whether dogs were offered (partly) several different baits? Or was there just one try per animal with one sort of bait? If I understand correctly from l 258 each dog was supposed to consume one bait of one sort only? L322ff (discussion) alludes to probable selection bias, so I would appreciate if it could be explained if any other possible bias could have happened and if any measures were taken to avoid these.

Discussion

L318 local bait acceptance: would you recommend then to pre-test with locally available products for other vaccination areas/countries?

Additionnally it would be good to have a broader outlook of the significance of this study for the planning of further vaccination campaigns (in Thailand but also internationally - are these results useful for other settings or in how far would they have to be repeated?) and a short estimation of average cost for these oral vaccination campaigns vs. CVR. (the benefit is to improve vaccination rates - what are the costs associated with these additional activities?) 

Author Response

Response to Reviewer 1 Comments

Point 1: Ll 44/45 provide reference.

Response: Sentence was slightly adapted so that it was aligned with the previous one and the same Reference could be used

Point 2: L 54 not sure why a median is presented here if a decreasing trend is to be shown.

Response: Agreed and the whole sentences was revised to “Also, in Thailand significant progress has been made in the elimination of rabies. Human rabies cases per year decreased from 200-300 in early 1980s [4] to three in 2019 (available at http://www.boe.moph.go.th/boedb/surdata/index.php). Animal rabies was also in the same trend; it was significantly reduced from 3-4,000 confirmed cases per year during 1997-2000 to 380 cases in 2019 (available at http://www.thairabies.net/trn/).”.

Point 3: L 58 how is this different from the median shown in l 54?

Response: It’s the number of rabies cases in dogs. For clarity it was revised to “In 2019, 87% of animal rabies in Thailand were reported in dogs (available at http://www.thairabies.net/trn/).”

Point 4: Information missing: how were the MDV campaigns organized: door to door? Central point vaccination?

Response: Extended the last sentence to cover recommended point. “…,which have been conducted annually during April-June by both methods, door-to-door and central point vaccination, depending on preference of local responsible authorities.”.

Point 5: More information on similar and distinctive characteristics of the municipalities with known problems reaching the stray dog population would be interesting. It could help identifying predictors of unaccessibility of dogs for future campaigns. However, I see this is not the purpose of this paper and I leave it up to authors

Response: We did not systematically explore this problem. We did interview local staffs and we found the common problem; restraining and vaccinating the free-roaming dog population. This was added in line 93.

Point 6: Fig.2: Baits are pictured very well, but could a bait in a sachet be shown, too? Otherwise to the unexperienced reader it is hard to understand what is meant by the "sachet".

Response: We added a picture of a sachet.

Point 7: L 173 provide reference concerning the mentioned application.
Response: We added the reference and revised the sequency accordingly (now line 183).

Point 8: L 188 "person offering the vaccine"
Response: Agreed and revised accordingly.

Point 9: Fig.3 Garbage dump area
Response 9: Agreed and revised the figure accordingly.

Point 10: L211 what is meant by "directly" opposed to "dropping in front of dog"?
Response: Word ‘directly’ in the abstract (L28) and material and method (L144) was deleted to avoid confusion. Direct means hand feeding, this description was inserted in line 200

Point 11: L219 ff it is not clear whether dogs were offered (partly) several different baits? Or was there just one try per animal with one sort of bait? If I understand correctly from L258 each dog was supposed to consume one bait of one sort only? L322ff (discussion) alludes to probable selection bias, so I would appreciate if it could be explained if any other possible bias could have happened and if any measures were taken to avoid these.
Response: We revised material and method (L 149) hopefully avoiding misunderstanding: “Only one type of bait type was used by a team during a particular day. In case of the egg bait, the team made a decision whether to add the cat liquid snack paste on the egg bait (egg+ bait) or not if the dogs seemed to be difficult to access.” This decision caused selection bias as mentioned in the discussion. The aim of the study was not to compare different bait type. We could not indentify other biases that could have influenced the study result.  

Point 12: L318 local bait acceptance: would you recommend then to pre-test with locally available products for other vaccination areas/countries?
Response: As mentioned in the last sentence of the discussion “Finally, the commercial egg-flavored bait was recommended during the post campaign meeting in order to avoid the risk of interrupting the cold-chain during preparation of the local-made intestine vaccine bait.” However, the acceptance of the commercial bait can potentially be further optimized by using local avaliable procuct such as cat liquid snack in this study. If bait acceptance of the commercial egg-flavored bait is not satisfactorily as tested under local conditions, additives should be pre-tested. Of course, these (local) additives may differ among countries (regions).

Point 13: Additionally, it would be good to have a broader outlook of the significance of this study for the planning of further vaccination campaigns (in Thailand but also internationally - are these results useful for other settings or in how far would they have to be repeated?) and a short estimation of average cost for these oral vaccination campaigns vs. CVR. (the benefit is to improve vaccination rates - what are the costs associated with these additional activities?) 
Response: ORV demonstrated many benefits to improve vaccination coverage in free-roaming dog population. Agreed that several (additional) studies should be done including aspects likes cost benefit, impact on herd immunity, frequency of ORV vaccination campaign. The discussion also addresses some of these aspects. However, it is often difficult to give a general indication of costs as they depend very much on the local situation. For example, see the extreme wide variation of costs per (parenterally) vaccinated dog during MDV campaigns in the published literature.

Reviewer 2 Report

The manuscript "Feasibility and effectiveness studies with oral vaccination of free-roaming dogs against rabies in Thailand" reports the findings of an ORV directed at an "inaccessible" dog population that could form the final barrier to elimination of dog rabies in countries that are attempting to eliminate the disease. The study is well presented and the statistical analysis is appropriate, the conclusions reflect the results of the study. There is only one major consideration for the authors, that is did the study result in a decline in rabies in the areas where the ORV was applied? As the study was conducted in 2020 it may be too early to know but the authors should address this issue in the conclusions as ultimately it is the only measure that counts.

Minor suggestions for the text:

Line 131. replace 'a short' with "brief"

Line 151. delete 'up'

Line 293. assume "expensive" not 'expansive'

Line 319. suggest "..to be kept at low temperature.."

Line 367. The authors use a lot of percentages throughout and this can be confusing, suggest clarification "Overall 79% of the inaccessible free-roaming dog population at the selected sites could be offered bait and 65.6% were subsequently considered successfully vaccinated." Presumably, the <65.6% is offset by the level of vaccination achieved in the "accessible" dog population so it would be of interest to speculate what the seroprevalence level might be in the dog population in totality.

Author Response

Response to Reviewer 2 Comments

Point 1: The manuscript "Feasibility and effectiveness studies with oral vaccination of free-roaming dogs against rabies in Thailand" reports the findings of an ORV directed at an "inaccessible" dog population that could form the final barrier to elimination of dog rabies in countries that are attempting to eliminate the disease. The study is well presented and the statistical analysis is appropriate, the conclusions reflect the results of the study. There is only one major consideration for the authors, that is did the study result in a decline in rabies in the areas where the ORV was applied? As the study was conducted in 2020 it may be too early to know but the authors should address this issue in the conclusions as ultimately it is the only measure that counts.

Response: The reviewer is correct that at the end a decline in the rabies incidence is the only ‘response’ that matters. However, it is extremely difficult to link the number of rabies cases with a specific control measure as so many other factors can play a role. For example, how to assess the influence of the present COVID-19 panendemic on rabies incidence and – surveillance? To assess the effect of ORV on the dog rabies incidence is a complicated topic, addressing it in detail in this paper would distract from the aim of this study.

Point 2: Line 131. replace 'a short' with "brief"

Response: Agreed and revised

Point 3: Line 151. delete 'up'

Response: Agreed and revised

Point 4: Line 293. assume "expensive" not 'expansive'

Response: agreed and revised.

Point 5: Line 319. suggest "..to be kept at low temperature.."

Response: agreed and revised.

Point 6: Line 367. The authors use a lot of percentages throughout and this can be confusing, suggest clarification "Overall 79% of the inaccessible free-roaming dog population at the selected sites could be offered bait and 65.6% were subsequently considered successfully vaccinated." Presumably, the <65.6% is offset by the level of vaccination achieved in the "accessible" dog population so it would be of interest to speculate what the seroprevalence level might be in the dog population in totality.

Response: Inaccessible means inaccessible for parenteral vaccination. We did not measure parenteral vaccination coverage. For clarification, revised the sentence to  

“Overall, 79.0% and 65.6% of the free-roaming dog population inaccessible for parenteral vaccination at the selected sites could be offered bait and were subsequently considered successfully vaccinated.”

Reviewer 3 Report

In this article entitled « Feasibility and Effectiveness Studies with oral vaccination of free-roaming dogs against rabies in Thailand, Karoon Chanchai and colleagues describe the design, logistics and results of an oral dog vaccination campaign performed in  5 remote sites in Thailand in march-august 2020. The goal of the study was to vaccinate a sub population of stray dogs, the free-roaming dogs without an owner, which escape the Mass Dog vaccination campaings  using  parenteral vaccines. The vaccine used was an attenuated  SPBN GASGAS rabies virus strain  prepared   by CEVA Innovations Center in Dessau, Germany.   Sachets filled with liquid vaccine (3ml  10e8.2 FFU/ml) were asociated with three types of baist , egg bait, egg+bait and pork intestine bait.

The experimenters carefully mapped and described the vaccination sites, noted the bait acceptance by the dogs, their preference for the bait, the way the bait was chewed, and fate of the vaccine sachet . For reasons of cost and the safety of the experimenters, blood samples were not taken. It is therefore impossible to know whether the dogs developed an antibody response and became immune. How the  criterion for the success of the vaccination was that the dog had punctured the sachet containing the vaccine is a valid surrogate. With this criterion they estimated that vaccination coverage was 65.6% , a value not different from parenteral  dog vaccination.

This carefully conducted and well thought study is a very instructive pilot experiment which should inspire new countries where rabies is endemic in stray dogs to organize this type of vaccination campaign.

The article is well written.

 A minor point to be improve dwould be the name of the baits: it is difficult to understand what represent "egg bait" and "egg + bait" (Fig 2). Besides, even the authors get lost, and use  other nomenclatures like "Egg" and “Egg +” (table 1), then in table 3 "egg-flavored +" and "Egg-flavored". Homogenization would be welcome.

Author Response

Response to Reviewer 3 Comments

Point 1: The experimenters carefully mapped and described the vaccination sites, noted the bait acceptance by the dogs, their preference for the bait, the way the bait was chewed, and fate of the vaccine sachet . For reasons of cost and the safety of the experimenters, blood samples were not taken. It is therefore impossible to know whether the dogs developed an antibody response and became immune. How the criterion for the success of the vaccination was that the dog had punctured the sachet containing the vaccine is a valid surrogate. With this criterion they estimated that vaccination coverage was 65.6%, a value not different from parenteral dog vaccination.
Response: The vaccination coverage of the targeted subpopulation (free-roaming dogs considered inaccessible for parenteral vaccination) was (by definition) 0% as these dogs cannot be restrained and handled during the traditional MDV-campaigns. Hence, reaching 65.6% vaccination coverage in these dogs that play a key role in rabies transmission is very important, esp. in countries like Thailand were these inaccessible dogs form a significant segment of the overall dog population.

Point 2: A minor point to be improved would be the name of the baits: it is difficult to understand what represent "egg bait" and "egg + bait" (Fig 2). Besides, even the authors get lost, and use  other nomenclatures like "Egg" and “Egg +” (table 1), then in table 3 "egg-flavored +" and "Egg-flavored". Homogenization would be welcome.
Response: We gave a definition of egg bait and egg+ bait in Line 116-122 (Material & Methods). Wording “egg bait’ and “egg+ bait” were used consistently after this throughout the manuscript. “A sachet filled with the liquid vaccine virus (3 ml, 10^8.2 FFU/ml) was incorporated in two different bait types, previously shown to be readily accepted by local free-roaming dogs in Thailand; an industrial manufactured egg-flavored bait (egg bait) and a locally produced intestine bait. As the study team observed that the acceptance of the egg bait could be further optimized, tuna- or chicken liver-flavored cat liquid snacks available in the local markets were pasted on one side of the outer surface of the bait immediately before offering the bait to the dog (egg+ bait)”. The manuscript has been checked once more, making sure these terms are used consistently.